Estuarine bivalve metabolic response mediated by environmental drivers

Lam-Gordillo Orlando orlando.lam-gordillo@niwa.co.nz
Douglas Emily J.
Hailes Sarah F.
Lohrer Andrew M.
National Institute of Water and Atmospheric Research , Hamilton , New Zealand
Waiho Khor
Electronic publication date: 2025 Nov 17
Publication date: 2025
Volume: 13
Electronic Location ID: e20357
Received 2025 Jul 28; Accepted 2025 Oct 16
Copyright: ©2025 Lam-Gordillo et al.
Copyright year: 2025
Copyright holder: Lam-Gordillo et al.
License: This is an open access article distributed under the terms of the Creative Commons Attribution License, which permits unrestricted use, distribution, reproduction and adaptation in any medium and for any purpose provided that it is properly attributed. For attribution, the original author(s), title, publication source (PeerJ) and either DOI or URL of the article must be cited.
License URL: https://creativecommons.org/licenses/by/4.0/

Keywords: Climate change, Environment, Stressors, Sediment, Metabolites, Bivalves

Funding: New Zealand Government’s Strategic Science Investment Fund (SSIF) to the National Institute for Water & Atmospheric Research NIWA; CEME2401/2501 The New Zealand Institute for Earth Science ESNZ; FPRS2604 This research was funded by the New Zealand Government’s Strategic Science Investment Fund (SSIF) to the National Institute for Water & Atmospheric Research (NIWA; CEME2401/2501) and The New Zealand Institute for Earth Science (ESNZ; FPRS2604). The funders had no role in study design, data collection and analysis, decision to publish, or preparation of the manuscript.

==============================
Humans are rapidly modifying environmental conditions in estuaries, which are among Earth’s most productive and dynamic ecosystems. Bivalve molluscs are key estuarine organisms, contributing to range of ecosystem functions and services, though human-induced environmental changes are affecting their behaviour, physiology, and fitness with implications at individual, population, community, and ecosystem levels. Understanding how estuarine bivalves respond and adapt to different environmental drivers will enable us to better predict change at multiple levels of biological organisation. In this study, we investigated the metabolites of a common and ecological important suspension-feeding bivalve in New Zealand, the cockle, Austrovenus stutchburyi. At seven of eight pre-established monitoring sites in a North Island estuary, we evaluated differences in cockle metabolite abundance, diversity, and composition, as well as relationships between cockle metabolites and environmental conditions. Our findings revealed differences in the abundance and diversity of cockle metabolites across sites, particularly in the metabolites alanine, aspartic acid, glutamic acid, glycine, proline, and succinic acid. The differences in metabolites across sites were mediated by the site-specific environmental conditions, in particular, the sediment’s mud content and organic matter. Differences in metabolites were most pronounced when comparing sites close to freshwater inputs versus sites located closer to the estuary mouth. In general, Austrovenus metabolite abundance was higher at sites with less signs of stress (i.e., close to the estuary mouth) and lower in sites with with higher mud content (i.e., close to freshwater inputs), while the metabolite diversity followed an inverse pattern. The metabolic responses of cockles appeared to be linked to processes such as feeding, oxygen regulation, and energy allocation. The observed metabolic trends highlight the complex interactions between cockles and their environment and provide insights into the metabolic responses of bivalves to the rapidly changing environment.

Introduction

Estuarine ecosystems are transitional zones where freshwater from rivers meets and mixes with saltwater from the ocean. Because of high nutrient and energy inputs from both endmembers, estuaries are biogeochemical hotspots and they are often noted for their productivity and biodiversity (Thrush et al., 2013; Douglas et al., 2018a; Rosenzweig et al., 2018; Douglas, Lohrer & Pilditch, 2019; Thrush et al., 2021a; Thrush et al., 2021b).

Environmental change driven by anthropogenic pressures is affecting estuarine ecosystems worldwide (IPCC, 2022; Deng, 2024; Prum, Harris & Gardner, 2024). Increases in the frequency and strength of stressors including pollution, habitat fragmentation, coastal development, and nutrient and sediment loads, exacerbated by climate change, are leading to profound alterations of the composition and structure of estuarine communities, with tangible effects on the processes, functions, and services these important ecosystems provide (Lohrer et al., 2010; Ellis et al., 2015; Hewitt, Ellis & Thrush, 2016; Thrush et al., 2017; Carrier-Belleau et al., 2021; Gammal et al., 2022; Douglas et al., 2023). Inputs of fine sediments (silt and clay) and nutrients from land are key stressors for estuaries which can cause shifts towards muddy, organic rich habitats that are prone to eutrophication (Thrush et al., 2003a; Lohrer, Hewitt & Thrush, 2006; Bierschenk, Savage & Matthaei, 2017). Sediment mud content is used as a measure of estuary degradation status since such changes in sedimentary environment are associated with reduced ecosystem functioning, changes in macrobenthic community composition and species loss (Thrush et al., 2004).

Bivalve molluscs are key components of estuarine ecosystems, with contributions to several ecosystem functions and services, including nutrient cycling, food provision, sediment turn-over, water filtration, and habitat provision (Harris et al., 2015; Hewitt, Ellis & Thrush, 2016; Ellis et al., 2017; Douglas et al., 2018a; Douglas, Lohrer & Pilditch, 2019; Rullens et al., 2019; Lam-Gordillo et al., 2024b). Suspension feeding bivalves exert top-down control on phytoplankton populations, affecting rates of nutrient cycling, contribute to benthic-pelagic coupling, and provide an important food resources for fish and birds (Thrush et al., 2003b; Newell, 2004; Thrush et al., 2006; Jones et al., 2011). Yet, increased sediment and nutrient loading to estuaries is challenging the survival and functioning of suspension feeding bivalves by altering primary production, seabed light and oxygen levels, and sediment characteristics such as grain size and organic matter content (Grant & Thorpe, 1991; Thrush et al., 2003b; Norkko, Hewitt & Thrush, 2006; Cummings et al., 2007).

Suspension feeding bivalves are particularly stressed by excessive turbidity and high sediment deposition rates (Thrush et al., 2003b; Norkko, Hewitt & Thrush, 2006; Jones et al., 2011; Lohrer et al., 2012). Their responses can be expressed at different levels, from altered individual feeding behaviour, physiology, and fitness over the short term, to changes in population dynamics due to differential growth, survival, and reproduction over the longer term (Norkko et al., 2002; Lohrer et al., 2006; Norkko, Hewitt & Thrush, 2006; Douglas et al., 2018a).

The bivalve Austrovenus stutchburyi (also known as cockle—hereafter Austrovenus), is a common native suspension feeder in New Zealand intertidal soft sediments (Jones et al., 2011; Ellis et al., 2015; Ellis et al., 2017; Douglas et al., 2023). Austrovenus plays an essential role in intertidal ecosystem functioning as it modifies sediment biogeochemistry (including nutrient regeneration and removal) and pore water solute exchanges through bioturbation and biodeposition (Lohrer et al., 2016; Douglas et al., 2018b; Tricklebank, Grace & Pilditch, 2021). Since large bioturbating fauna accelerate the nutrient removal ecosystem service (i.e., denitrification) (Webb & Eyre, 2004; Stief, 2013), Austrovenus are expected to be important for ecosystem resilience to eutrophication. Austrovenus are also culturally important as a food resources for indigenous Māori, and economically important for recreational and commercial harvesters (Marsden & Adkins, 2010; Fenton et al., 2024; Lam-Gordillo et al., 2024a; Yeoh et al., 2024). However, there is increasing evidence that the populations of Austrovenus are declining in several areas of New Zealand, mainly due to increases in bed sediment muddiness, suspended solids in the water column, increasing air and water temperatures, and more frequent heatwaves (Lohrer et al., 2004; Thrush et al., 2004; Tricklebank, Grace & Pilditch, 2021; Douglas et al., 2023; Lam-Gordillo et al., 2024b).

Although there is a wealth of research concerning the effects of environmental conditions on suspension feeding bivalves (e.g., Norkko et al., 2002; Lohrer et al., 2006; Norkko, Hewitt & Thrush, 2006; Douglas et al., 2018a), there is little information about the metabolic response of estuarine suspension feeding bivalves to human-modified environmental drivers. This limits our understanding of how estuarine organisms respond and adapt to different environmental drivers, and our ability to predict community change and the effects on estuarine ecosystem functioning and service provision.

In this study, we surveyed an estuary in New Zealand with the focus of investigating the metabolic response of Austrovenus to site-specific variation in environmental conditions. Specifically, we aimed to evaluate differences in Austrovenus metabolic abundance, diversity, and composition across several monitoring sites within the estuary and investigate the relationship between Austrovenus metabolites and the environmental conditions of the estuary. We hypothesised that higher metabolite abundance and diversity will be recorded at sites with lower levels of degradation/stress (e.g., low mud and organic content in sediment).

Methods

Study area

The investigation was carried out in Waihı¯ Estuary in the Bay of Plenty region, North Island, New Zealand (−37.7695, 176.4870; Fig. 1). Waihı¯ Estuary is a tidal lagoon type estuary that is mostly enclosed by a coastal sandspit but permanently open to the sea at its narrow mouth. It is dominated by intertidal soft-sediment flats and has a relatively high freshwater inflow volume for an estuary of its size (Lam-Gordillo et al., 2024a). Eight sites had been sampled previously for estuarine ecological health assessments (Lam-Gordillo et al., 2024a). We surveyed the eight pre-established monitoring sites, however, Austrovenus was only presented and collected at seven of the eight sites. These seven sites dispersed across the estuary were the focus of the current study. Environmental data and biological data were collected in the same survey and paired at the same locations.

Figure 1 Map of Waihı¯ Estuary, New Zealand.

The eight pre-established monitoring sites (Lam-Gordillo et al., 2024a) are showed. Austrovenus sampling occurred in all sites except site 6 (where it was not present in sufficient abundance). Blue lines show the local freshwater inputs. Imagery retrieved from Land Information New Zealand (https://data.linz.govt.nz/).

Data collection

Environmental data

To investigate the metabolic response of Austrovenus to environmental conditions, we selected three sedimentary variables as potential influencers of metabolic change in Austrovenus. Chlorophyll-a (Chl a; μg/g) content in sediment, sediment organic matter content (OM; %), and sediment grain size (% dry weight) were selected due to their potential to change in response to freshwater (i.e., nutrients, sediment) loadings and influence estuarine communities.

Sediment samples were collected as previously described in Lam-Gordillo et al. (2024a). Particularly, sediment samples for Chl a, sediment grain size, and OM were collected at each of the seven sampling sites. Using a small PCV corer with dimensions of 26 mm diameter and 20 mm deep (Lam-Gordillo et al., 2024a). We collected three replicate cores of the upper 0–2 cm of the sediment (Lam-Gordillo et al., 2024a). Sediment from these three cores was combined for each replicate to capture localized variation (Lam-Gordillo et al., 2024a). Prior to analysis, each small core was homogenized and sub-sampled for analysis of Chl a, OM, and grain size (Lam-Gordillo et al., 2024a). Following regional standardised protocols previously described by Drylie (2021) and Lam-Gordillo et al. (2024a), Chl a was extracted from freeze dried sediments by boiling (80 °C) in 90% ethanol. The extract was measured spectrophotometrically, and an acidification step was included to separate degradation products (phaeophytin) from Chl a (Sartory, 1982). For sediment grain size, samples were homogenised and then digested in ∼9% hydrogen peroxide until frothing ceased. Samples were wet sieved through 2,000 µm, 500 µm, 250 µm, 125 µm, and 63 µm mesh sieves. Pipette analysis was used to separate the <63 µm fraction into >3.9 µm and ≤3.9 µm. All fractions were then dried at 60 °C until a constant weight is achieved (fractions are weighed at ∼40 h and then again at 48 h) to obtain the percentage weight of gravel/shell hash (>2000 µm), coarse sand (500–2,000 µm), medium sand (250–500 µm), fine sand (125–250 µm), very fine sand (62.5–125 µm), silt (3.9–62.5 µm) and clay (≤3.9 µm). Mud was defined as the sum of the silt and clay fractions (Lam-Gordillo et al., 2024a). Organic matter content was determined by drying the sediment at 60 °C for 48 h and then combusting it at 400 °C for 5.5 h, with OM expressed as percent dry weight lost on ignition (Lam-Gordillo et al., 2024a).

Biological data

At each of the seven sampling sites, 50 individual Austrovenus were collected by finger ploughing the upper 0–5 cm of the sediment column. Collection of bivalves was carried out in Austral Summer (February 2024) coinciding with the reproductive season of Austrovenus (McKinnon, 1996; Jones, 2011; Adkins, Marsden & Pirker, 2016). We ensured that all specimens used were fully mature, based on their body size (≥30 mm shell length), which is consistent with established size thresholds for sexual maturity in this species. All Austrovenus were dissected immediately after collection (within 2 h). For each site, muscle tissue from the 50 individuals were amalgamated into five samples (i.e., totalling–35 samples from seven sites). Samples were snap frozen and stored in liquid N2 in the field and for transport. In the laboratory, samples were stored frozen at −40 °C degrees until metabolomic processing.

Metabolomic data

We followed the protocols described by Smart et al. (2010) and Lam-Gordillo et al. (2025) for the analysis of metabolites in muscle tissue of Austrovenus using a methyl chloroformate (MFC) alkylation derivatisation method. This method has proven useful for derivatising amino and non-amino organic acids, and some primary amines and alcohols (Smart et al., 2010; Lam-Gordillo et al., 2025). Samples of Austrovenus muscle tissue were processed following the same methodology as described in Lam-Gordillo et al. (2025), first samples were freeze-dried using a refrigerated vapor trap SpeedVac concentrator (Thermo Scientific, Waltham, MA, USA). Metabolite extraction was performed for each of the 35 samples by adding a cold 1:1 methanol:water solution to the dried samples (Lam-Gordillo et al., 2025). Samples were vortexed and centrifuged, and the supernatants (derivatized samples) were then analysed with a gas chromatograph GC7890B coupled to a quadruple mass spectrometer MSD5977A (Agilent Technologies, Santa Clara, CA, USA) (Lam-Gordillo et al., 2025). Different types of quality controls (QC) were used to ensure reproducibility of MFC measurements, which included blank samples, amino acid mixtures, and pooled QC samples from all samples (Smart et al., 2010; Lam-Gordillo et al., 2025). The analysis of raw spectral data, data mining, and metabolite identification was performed using Chem Station (Agilent Technologies, Inc., Santa Clara, CA, USA), Automated Mass Spectral Deconvolution and Identification System (AMDIS) software (http://www.amdis.net), and MassOmics R package—The University of Auckland (Guo et al., 2020; Lam-Gordillo et al., 2025). Metabolomic data was normalised to biomass (based on the mean bivalve tissue weight analysed) and the internal standard to compensate for potential technical variations prior to data analyses (Lam-Gordillo et al., 2025). MFC analysis, data mining, and metabolite identification were performed by the Mass Spectrometry Centre, The University of Auckland, Auckland, New Zealand (Lam-Gordillo et al., 2025).

Data analysis

We used three ecological metrics to analyse Austrovenus metabolomic patterns across sampling sites in Waihı¯ Estuary. (1) Metabolite abundance: total biomass of each metabolite found at each sampling site, (2) Metabolite diversity: expressed as Shannon-Weiner (H’) diversity which accounts for both richness (number of different metabolites) and evenness (how evenly individuals are distributed among metabolites, and (3) metabolite structure: the composition and arrangement of metabolites within a site, including how metabolites interact and how their abundances are distributed.

We test for differences in Austrovenus metabolomic abundance (metabolite biomass) and metabolomic diversity (Shannon-Weiner H’) between sampling sites using univariate PERMutational ANalysis Of VAriance (PERMANOVA) tests based on Euclidean distance for the single variables, permutation of residuals under a reduced model, sums of squares type III, and 9999 permutations. Multiple pair-wise tests with 9999 permutations were conducted if the fixed factor (Site) was significant to identify which groupings contributed to differences from PERMANOVA main tests. To facilitate the differences visualisation, Austrovenus metabolomic abundance and diversity data were presented as boxplots constructed using the package ‘ggpubr’ (Kassambara, 2020) in R software (R Core Team, 2022).

To assess metabolomic structure and investigate differences between sites, we created a bootstrapped non-Metric Multidimensional Scaling (nMDS) ordination plot based on Bray-Curtis similarities (Anderson, Gorley & Clarke, 2008). To test for differences in metabolomic structure between sites, a PERMANOVA test was performed using Bray Curtis similarities, permutation of residuals under a reduced model, sums of squares type III and 9,999 permutations (Anderson, Gorley & Clarke, 2008). Multiple pair-wise tests with 9,999 permutations were conducted if the fixed factors were significant to identify which groupings contributed to differences from PERMANOVA main tests.

A SIMilarity PERcentage breakdown (SIMPER) analysis (Anderson, Gorley & Clarke, 2008) was carried out to further explore the main metabolites accounting for >70% of the dissimilarity between sites. The metabolites identified by the SIMPER analyses were further tested for differences between sites following the same methodology as previously described. A waffle plot was used to visualise the SIMPER analysis results.

The relationship between Austrovenus metabolomic expression and environmental conditions was also investigated using distance-based ReDundancy Analysis (dbRDA) (McArdle & Anderson, 2001). Additionally, the individual relationships between metabolite ecological metrics (abundance and diversity), key metabolites (alanine, aspartic acid, glutamic acid, glycine, proline, and succinic acid) and sediment characteristics (e.g., Mud, OM, and Chl a) were evaluated using Pearson correlations and visually plotted using the package “corrplot” (Wei et al., 2017) in R software. PERMANOVAs, bootstrap nMDS, SIMPER, and dbRDA analysis were carried out using PRIMER v7 with PERMANOVA add on.

Results

Environmental conditions

Environmental conditions varied across the sampling sites (Table 1). In general, most of the sites were characterised by medium and fine sediments except site 7 and 8 (Table 1). Site 1 showed the highest content of fine sand in sediment (53%), followed by Site 3 (40%), while sediment at Site 5 had the highest medium sand content (Table 1). Site 7 was the muddiest site (57% mud). Site 8 also showed high sediment mud content (36%), high fine sand content (30%), and the highest OM in sediment (Table 1). Site 2 showed the second highest OM content in sediment, while the lowest OM content was identified in Site 1 (Table 1). The highest concentration of Chl a was found in Site 2 (49 μg/g) and the lowest concentration in Site 7 (15 μg/g).

Table 1 Summary of the sedimentary conditions.

Sedimentary conditions were recorded across the seven sampling sites in Waihı¯ Estuary.

Sites	Gravel (%)	Coarse Sand (%)	Medium Sand (%)	Fine Sand (%)	Very Fine Sand (%)	Silt (%)	Clay (%)	Mud (%)	OM (%)	Chl a (µg/g)	
1	0.08	3.14	21.81	53.36	15.61	4.22	1.78	6.00	1.41	29.20	
2	1.06	0.83	2.59	31.01	39.93	21.31	3.28	24.58	3.01	49.06	
3	0.12	4.35	23.38	39.82	20.42	10.17	1.74	11.91	2.25	33.69	
4	0.02	3.76	15.27	29.91	29.70	19.60	1.73	21.33	2.41	25.24	
5	0.09	8.82	33.20	20.90	12.24	22.95	1.80	24.75	2.04	18.65	
7	0.02	4.58	14.73	13.60	9.58	54.37	3.12	57.49	2.93	15.13	
8	0.79	5.84	8.66	18.97	29.76	32.67	3.31	35.98	3.53	39.97	

Metabolomic patterns of Austrovenus bivalves

In total, 47 Austrovenus metabolites were identified across the sampling sites (Table S1). Austrovenus metabolite abundance and diversity were different across sampling sites (Fig. 2). The highest mean abundance of metabolites was found at Site 1, followed by Site 4 and Site 3, while the lowest mean metabolite abundance was identified in Site 7 (Fig. 2A). There were significant differences in metabolite abundance between sites (PERMANOVA p < 0.05; Fig. 2A). Pairwise tests revealed that the abundance of metabolites at Site 7 was significantly different to all other sites (p < 0.05; Table 2; Fig. 2A). Metabolite abundance at Site 1 was significantly different to sites 2, 3, and 5. Significant differences between Site 2 and Site 3, and between Site 3 and Site 5, were also recorded (p < 0.05; Table 2; Fig. 2A).

Figure 2 Boxplots showing the metabolites recorded across the sampling sites in Waihı¯ Estuary.

(A) Metabolites abundance (metabolites biomass) and (B) metabolites diversity (Shannon-Weiner–H’) recorded across the seven study sites. Thick lines = median, dots = outliers.

Table 2 Summary of PERMANOVA pair-wise test.

Comparisons of the abundance of metabolites recorded across sites. Significant differences are shown in bold.

Groups	t	P(perm)	Unique perms	
1, 2	5.568	0.008	126	
1, 3	3.179	0.031	126	
1, 4	0.647	0.564	126	
1, 5	4.839	0.008	126	
1, 7	10.948	0.006	126	
1, 8	2.025	0.070	126	
2, 3	4.190	0.008	126	
2, 4	2.349	0.051	126	
2, 5	0.636	0.492	126	
2, 7	6.403	0.008	126	
2, 8	1.129	0.300	126	
3, 4	0.674	0.527	126	
3, 5	3.247	0.007	126	
3, 7	10.588	0.009	126	
3, 8	0.732	0.538	126	
4, 5	1.988	0.083	126	
4, 7	6.191	0.010	126	
4, 8	1.005	0.364	126	
5, 7	6.858	0.009	126	
5, 8	0.756	0.409	126	
7, 8	5.223	0.007	126	

Diversity of metabolites in Austrovenus significantly varied across sites (PERMANOVA p < 0.05; Fig. 2B). The highest mean diversity was identified at Site 7, followed by Site 8, while the lowest diversity of metabolites was identified at Site 4 (Fig. 2B). Pairwise tests revealed that the diversity of metabolites found at Site 7 was significantly different to all other sites except Site 8 (p < 0.05; Table 3; Fig. 2B). Diversity of metabolites at Sites 3 and 4 were also significantly different to Sites 5, 7, and 8, with Site 4 also significantly different than Sites 1 and 2 (p < 0.05; Table 3; Fig. 2B).

Table 3 Summary of PERMANOVA pair-wise test.

Comparisons of the diversity of metabolites recorded across sites. Significant differences are shown in bold.

Groups	t	P(perm)	Unique perms	
1, 2	0.558	0.592	126	
1, 3	2.314	0.056	126	
1, 4	3.189	0.008	125	
1, 5	0.342	0.742	126	
1, 7	2.812	0.030	126	
1, 8	1.555	0.162	126	
2, 3	1.471	0.195	126	
2, 4	2.535	0.025	126	
2, 5	0.422	0.701	126	
2, 7	2.970	0.040	126	
2, 8	1.824	0.131	125	
3, 4	1.439	0.198	126	
3, 5	2.570	0.016	126	
3, 7	3.979	0.008	126	
3, 8	3.285	0.009	126	
4, 5	3.332	0.007	126	
4, 7	4.544	0.007	126	
4, 8	3.938	0.009	126	
5, 7	3.117	0.008	126	
5, 8	2.038	0.016	126	
7, 8	1.658	0.139	126	

Significant differences in the structural composition of metabolites within Austrovenus tissues were identified across sites (PERMANOVA p < 0.05; Fig. 3). The bootstrapped nMDS plot showed a clear differentiation between the composition of metabolites in Site 7 and all the other sites (Fig. 3). Pairwise test revealed differences in metabolite composition at some but not all sites, e.g., Site 8 did not differ from Sites 2, 3, and 4, and Site 3 did not differ from Site 4 (Table 4; Fig. 3).

Figure 3 Bootstrapped non-Metric Multidimensional Scaling (nMDS) plot.

The plot is showing the metabolomic structure across the study sites in Waihı¯ Estuary.

Table 4 Summary of PERMANOVA pair-wise test.

Comparisons of the structure composition of metabolites recorded across sites. Significant differences are shown in bold

Groups	t	P(perm)	Unique perms	
1, 2	3.554	0.008	126	
1, 3	1.835	0.025	126	
1, 4	1.405	0.121	126	
1, 5	4.834	0.008	126	
1, 7	6.363	0.010	126	
1, 8	1.971	0.030	126	
2, 3	2.457	0.008	126	
2, 4	2.127	0.040	126	
2, 5	3.172	0.009	126	
2, 7	4.574	0.008	126	
2, 8	1.222	0.209	126	
3, 4	0.858	0.551	126	
3, 5	4.164	0.009	126	
3, 7	5.849	0.007	126	
3, 8	1.481	0.063	126	
4, 5	3.004	0.008	126	
4, 7	5.213	0.008	126	
4, 8	1.696	0.062	126	
5, 7	4.760	0.007	126	
5, 8	2.428	0.009	126	
7, 8	4.633	0.007	126	

The SIMPER analysis identified six metabolites as the drivers of the significant differences across sites: alanine, aspartic acid, glutamic acid, glycine, proline, and succinic acid. Alanine and glycine were the metabolites that were the most consistently present and abundant across the sites, while glutamic acid and proline were found only in sites 5 and 7 (Fig. 4). Significant differences in abundance of metabolites were identified across all sites (PERMANOVA p < 0.05; Table S2; Fig. 5). The abundance of all metabolites at Site 7 was reduced relative to all other sites (Fig. 5). Alanine was significantly more abundant at Site 4 and significantly lower at Site 7 (p < 0.05; Table S2; Fig. 5A). Glycine showed a similar pattern as alanine: significantly higher at Site 4 and significantly lower at Site 7 (p < 0.05; Table S2; Fig. 5B). Aspartic acid, succinic acid, proline, and glutamic acid were all significantly higher at Site 1 and significantly lower at Site 7 (p < 0.05; Table S2; Figs. 5C and 5F).

Figure 4 Waffle plot showing a summary of the SIMPER analysis.

Metabolomic differences between sites in Waihı¯ Estuary. Each tile in the waffle plot represents a percentage contribution (1%) to the total dissimilarity.

Figure 5 Boxplots showing the abundance (biomass) of the most relevant metabolites identified by the SIMPER analysis across sampling sites in Waihı¯ Estuary.

(A) Alanine, (B) Glycine, (C) Aspartic acid, (D) Succinic acid, (E) Proline, and (F) Glutamic acid. Thick lines = median, dots = outliers.

The Distance based ReDundancy Analysis (dbRDA) explained a total of 93% of the total variation in metabolites structure (Fig. 6). The analysis indicated strong correlations between Austrovenus metabolite composition and environmental conditions (i.e., coarse, medium, fine, and very fine sand, silt, Mud, and Chl a concentrations). The dbRDA plot also revealed that Austrovenus metabolites at Site 1 were correlated with coarse sand, and with sediment mud content at Site 8, while sites 2, 3, and 4 were less correlated to mud (Fig. 6). Some individual Austrovenus metabolites were correlated with environmental conditions, e.g., alanine and glycine were correlated with coarse sand, and aspartic acid with fine sand, very fine sand, and Chl a (Fig. 6).

Figure 6 Distance-based ReDundancy Analysis (dbRDA) correlating Austrovenus metabolites with sediment conditions in Waihı¯ Estuary.

Two overlays are presented: sedimentary conditions (Sediment grain sizes, Chla: Chlorophyll a, OM: Sediment organic matter content, Mud: Mud content in sediment) are shown in black, metabolites in blue.

Pearson correlation showed significant correlations between metabolite abundance and mud and OM, while no significant correlations were found between metabolite diversity and the sedimentary characteristics (Fig. 7). Metabolite abundance was significantly positive correlated to mud, while significatively negative correlate to OM (Fig. 7). The key metabolite identified by the SIMPER analyses also significatively correlated with the sedimentary characteristics, except proline (Fig. 7). Alanine and glycine were significantly positive correlated to mud and significantly negative correlated to OM, aspartic acid and glutamic acid were significantly positive correlated to mud and Chl a, while succinic acid was only significantly positive correlated to Chl a (Fig. 7).

Figure 7 Pearson correlation between Austrovenus. metabolite ecological metrics, key individual metabolites and sediment conditions in Waihı¯ Estuary.

Colour scale showed the strength of the correlation, blue colours show positive correlations, red colours showed negative correlations, and * showed the significant (p < 0.05) correlations. N, Metabolite abundance; H, Metabolite diversity; AC, Aspartic acid; Al, Alanine; GA, Glutamic acid; Gl, Glycine; Pr, Proline; SA, Succinic acid; Mud, mud content in sediment; OM, organic matter content in sediment; Chla, Chlorophyll a.

Discussion

Organisms living in estuaries are influenced by a range of human-induced pressures. Changes in environmental conditions are driving alterations of the behaviour, physiology, and fitness of estuarine fauna with implications for ecological functioning at individual, community, and ecosystem levels. Here, we investigated the metabolic expression of a suspension feeding bivalve—Austrovenus stutchburyi across sites that spanned a gradient of environmental degradation and identified relationships between Austrovenus metabolites and sedimentary environmental conditions.

In accordance with our hypothesis, the abundance of Austrovenus metabolites was higher at sites with less signs of stress (i.e., mud and organic content) in Waihı¯ Estuary, yet diversity of Austrovenus metabolites was higher at sites with higher mud content. The metabolic processes of bivalves can be significantly influenced by environmental conditions (e.g., temperature, nutrients, salinity, pollutants, oxygen, sediment), which leads to variations in their metabolic pathways and thus the abundance and diversity of metabolites expressed (Smyth et al., 2018; Steeves et al., 2018; Saulsbury et al., 2019; Jiang et al., 2020; Georgoulis et al., 2022; Venter et al., 2023; Venter et al., 2024). In our study, differences in metabolite abundance and diversity were more evident between sites closer to freshwater inputs and sites in the upper and middle sections of the estuary. Previous studies have shown that macrobenthic communities, including suspension feeding bivalves, are structured by water column characteristics (e.g., food supply, temperature, salinity, and dissolved oxygen) and the sedimentary environment in which they burrow (e.g., mud and organic matter content, which influence permeability and biogeochemistry) (e.g., Lohrer et al., 2004; Norkko, Hewitt & Thrush, 2006; Douglas et al., 2018a; Douglas, Lohrer & Pilditch, 2019; Villnäs et al. 2019). Austrovenus at Site 7, farthest from the mouth of the estuary, showed the lowest abundance of metabolites but the highest metabolite diversity. Sites1, 2, and 3, which were closer to the mouth and more regularly flushed with clean seawater, exhibited the opposite pattern. These findings were aligned with sedimentary conditions: Site 7 with relatively high mud and low Chl a, and Sites 1–3 characterised by less mud and higher Chl a, showing how metabolite expression correlated with site-specific environmental conditions.

Bivalves, especially suspension feeders, are affected by high sediment loads such as increased mud content in sediment and high levels of suspended sediments in water (Lohrer et al., 2004; Lohrer et al., 2006; Norkko, Hewitt & Thrush, 2006; Jones et al., 2011; Lohrer et al., 2016; Douglas et al., 2018a). High content of mud in sediment can impact bivalve metabolism by altering the availability of food and oxygen. Changes in bivalves’ metabolic expression and processes could be influenced by reduced feeding efficiency and depletion of oxygen, potentially increasing the production of anaerobic metabolites due to lower oxygen levels in the sediment (De la Huz, Lastra & López, 2022; Smyth et al., 2018; Steeves et al., 2018; Saulsbury et al., 2019). High levels of suspended sediment in the water column are thought to affect bivalve feeding and respiration by clogging their gills and reducing their ability to filter particles and breathe efficiently (Norkko, Hewitt & Thrush, 2006; Lohrer et al., 2016; Smyth et al., 2018; Steeves et al., 2018; Saulsbury et al., 2019). This could lead to increased production of stress-related metabolites associated with shifts towards anaerobic metabolism (De la Huz, Lastra & López, 2022; Smyth et al., 2018; Steeves et al., 2018; Saulsbury et al., 2019). In low oxygen conditions, some bivalves are able to shift from aerobic to anaerobic metabolic pathways, resulting in the accumulation of anaerobic metabolites and triggering the production of stress-related metabolites that help bivalves survive in such conditions (Alfaro, Nguyen & Mellow, 2019; Jiang et al., 2020; Georgoulis et al., 2022; Muznebin et al., 2022; Hu et al., 2023; Venter et al., 2023). We found evidence of different metabolites across sampling sites, with alanine, aspartic acid, glutamic acid, glycine, proline, and succinic acid as the six most relevant metabolites. Alanine and succinic acid are metabolites related to shifts from aerobic to anaerobic respiration and are also indicative of metabolic stress, signalling a reduced efficiency in energy production (Ivanina et al., 2013; Jiang et al., 2020; Hu et al., 2023; Venter et al., 2023). Aspartic acid and glutamic acid are related to the tricarboxylic acid (TCA) cycle, which supports increased energy demands when the environmental conditions are not optimal (Muznebin et al., 2022; Venter et al., 2023), while metabolites such as glycine and proline respond to heat stress, stabilizing proteins and membranes, and helping cells maintain their structure and function under thermal stress (Muznebin et al., 2022; Hu et al., 2023; Venter et al., 2024; Lam-Gordillo et al., 2025). Interestingly, the six main metabolites, and in general the total abundance of metabolites, were significantly lower in Site 7 compared to any other site, which could be an adaptative response of Austrovenus to the sedimentary conditions at this site. The increased synthesis of these metabolites in response to high mud content in sediment could reflect adaptative mechanisms focused on enhancing cellular protection, shifts in respiration pathways, maintenance of homeostasis, and survival (Alfaro, Nguyen & Mellow, 2019; Jiang et al., 2020; Georgoulis et al., 2022; Muznebin et al., 2022; Hu et al., 2023; Venter et al., 2023; Lam-Gordillo et al., 2025). Decreases of metabolites at Site 7 may indicate increased use of these metabolites to prevent cumulative stress, cellular malfunction, and maintain stability and cell structure (Nguyen et al., 2018; Alfaro, Nguyen & Mellow, 2019; Liu et al., 2023; Venter et al., 2023; Venter et al., 2024; Lam-Gordillo et al., 2025), while increased metabolite abundance at Sites 1, 2, 3, 4, and 8 may indicate a stockpiling of these metabolite at places where environmental conditions are less stressful.

Our study revealed differences in Austrovenus metabolites across the seven sampling sites in Waihı¯ Estuary as well as strong correlations between Austrovenus metabolites and sedimentary conditions (i.e., mud and organic matter content in sediment). While sediment characteristics were considered a key explanatory factor, it is possible that other environmental factors and bivalve physiological characteristics contributed to the observed metabolic variation. Previous research has demonstrated that bivalve metabolites are also shaped by a combination of other environmental drivers, including water column properties such as temperature, salinity, and dissolved oxygen, microbial interactions, life history traits, and physiological stressors, which strongly influence amino acid metabolism, energy allocation, and stress response (Viant, 2007; Viant et al., 2009; Ellis et al., 2014; Bayona, De Voogd & Choi, 2022; Lam-Gordillo et al., 2025). Thus, we suggest the implementation of more holistic approaches that integrates multiple variables to disentangle the complex interactions shaping bivalve metabolic variation.

Furthermore, it is important to note that the results presented in this study reflect a narrow temporal window and may not capture the full extent of seasonal variability. Environmental conditions, biological activity, and sedimentary processes can fluctuate significantly throughout the year, potentially influencing the metabolite variation. While our findings offer valuable insights into the system during the sampling period, we acknowledge this limitation and recommend that future research incorporate year-round or long-term sampling to get a more holistic picture of the metabolite dynamics and improve the generalizability of the results.

Conclusion

This study elucidates the metabolic changes of Austrovenus stutchburyi across sites spanning a gradient in degradation status in Waihı¯ Estuary. Abundance and diversity of Austrovenus metabolites varied across sites, showing the greatest differences between sites closer to the freshwater inputs (the most impacted sites) and sites located closer to the estuary mouth (most frequently flushed with cleaner coastal ocean water). In general, metabolite abundance was higher at sites with less mud content and lower in sites with higher mud content in sediment, while metabolite diversity was higher at sites with higher mud content and lower in sites with lower mud content in sediment. We identified six key metabolites Alanine, Aspartic acid, Glutamic acid, Glycine, Proline, and Succinic acid, as the main drivers of differences across sites. Our findings also revealed that the differences in Austrovenus metabolite abundance and composition were correlated with the site-specific environmental conditions, but mainly driven by the muddiness of the sediment and the organic matter content in sediment. We provide evidence showing that changes in metabolite abundance, diversity, and composition responded to processes such as feeding efficiency, oxygen regulation, and energy demands. The observed metabolic patterns highlight the complex interactions between suspension feeding bivalves and their environment, emphasizing the importance of understanding these metabolic responses for assessing the health and resilience of bivalve populations in changing ecosystems. Further research in this area can provide valuable insights into the adaptive mechanisms of bivalves and inform conservation and management strategies for these important aquatic organisms.

Supplemental Information

Supplemental Information 1 Supplementary tables

Supplemental Information 2 Austrovenus metabolic data collected in Waihı¯ Estuary (Raw data)

Raw data of the metabolites identified in Austrovenus across the seven sampling sites in Waihı¯ Estuary.

We would like to thank Te Rūnanga o Ngāti Whakahemo and Professor Kura Paul-Burke (Waikato University) for inviting and warmly welcoming the NIWA team on to the Pukehina Marae and provide access to carry out the fieldwork and experimentation in Waihı¯ Estuary. The authors also thank Saras Green and Jin Wang the Mass Spectrometry Centre, The University of Auckland, Auckland, New Zealand for assistance with MFC analysis, data mining, and metabolite identification. This research was part of the Ecosystem Function and Health project and Ecosystem resilience and rehabilitation in a changing climate in NIWA’s Coasts & Estuaries Centre.

Additional Information and Declarations

Competing Interests

Author Contributions

Data Availability

The authors declare there are no competing interests.

Orlando Lam-Gordillo conceived and designed the experiments, performed the experiments, analyzed the data, prepared figures and/or tables, authored or reviewed drafts of the article, and approved the final draft.

Emily J. Douglas conceived and designed the experiments, performed the experiments, authored or reviewed drafts of the article, and approved the final draft.

Sarah F. Hailes performed the experiments, authored or reviewed drafts of the article, and approved the final draft.

Andrew M. Lohrer conceived and designed the experiments, performed the experiments, prepared figures and/or tables, authored or reviewed drafts of the article, and approved the final draft.

The following information was supplied regarding data availability:

The data used is available in the Supplementary File.

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
