# Peer review of "Estuarine bivalve metabolic response mediated by environmental drivers"

_PeerJ, doi:10.7717/peerj.20357_

## Round 0.1 · original submission · Major Revisions

· Academic Editor

Major Revisions

I agree with both reviewers that albeit scientifically sound and well written, several aspects of the manuscript needs further attention, especially in clarifying biological context, environmental variability, and analytical interpretation, to strengthen its conclusions and readability.

Reviewer 1 ·

Basic reporting

Orlando Lam-Gordillo and co-authors have studied how estuarine shellfish metabolism is affected by different environmental conditions. They carried out this study at eight different sites. Their experiment is well replicated, and the topic is new and important. Although the manuscript is generally well-written, it can still benefit from the following suggestions. Authors have used ecological community analysis to describe metabolite results, but the manuscript needs several improvements before it is ready for publication.

Specific Comments:

Line 16: The authors mention seven experimental sites, but the figure shows eight. Please check and make this consistent.

Line 21: Since the study focuses on metabolites, the authors should mention the main types of metabolites found and explain which were increased or decreased. This will help future studies to focus on important ones.

Line 116: Please explain clearly from where the sediment samples were collected. I assume from those 7–8 sites, but it is not clearly mentioned. Also, give more detail on the number of samples, the amount of each sample, when they were collected, and how many replicates per site. Right now, it only says how the samples were processed (sieving, measurements), not how they were collected.

Line 140: Were the samples stored in a special way or placed in special containers before processing? This is important, because metabolite levels can change during storage.

Line 152: Were the samples freeze-dried? Please clarify this.

Line 182: Please mention in the abstract that you analyzed metabolite composition using ecological community metrics like Bray-Curtis similarity.

Line 201: Please confirm in the methods if the Austrovenus bivalves and sediment samples were collected from the same locations.

Line 217: Are you reporting total metabolite abundance or relative composition? Please explain if you mean there was more or less of something, or just different types.

Line 221: The term “Metabolite Diversity” is not very clear. Do you mean the number of different metabolites (richness) or their evenness (relative abundance)? Since you are using ecological approaches, you may want to use terms like “metabolite alpha diversity” or “beta diversity” to make your analysis easier to understand.

Lines 234–244: Some of these important results should also be included in the abstract to show key findings.

Line 268: It would be helpful to test if environmental variables are related to metabolite changes. Please consider adding correlation analysis between environmental parameters and metabolite data. You can check microbiome studies that use ecological methods to show relationships between diversity and environment.

Line 335: The term “metabolite expression” may not be correct in this context. A better term could be “metabolite content” or “metabolite levels.”

Conclusion: The conclusion is too general. Please summarize the main results and clearly explain how metabolites are linked to environmental factors—either by site or averaged across all sites. This will help make the study more useful for other researchers.

Figures: Data presentation in the figures is clear and well-organized. Good work in this area.

Experimental design

See Above

Validity of the findings

See Above

Additional comments

See Above

Reviewer 2 ·

Basic reporting

The impact and relationship of the environment was assessed from the point of view of metabolite composition in the cockle Austrovenus stutchury sampled from seven different sites of an estuary at New Zealand. The MS is well written in most of the parts, I recommend some minor changes in order to improve the readability. The topic will be quite interesting for the readers of this journal. Although the results are well interpreted, there are some issues that were not considered in this study. The authors mentioned that several metabolites were measured in muscle samples. Only six metabolites were significant different among sampled sites. Mostly of them are aminoacids and one dicarboxilic acid (i.e. succinic acid). Although a plausible explanation was stated, in accordance with environmental factors assessed, is quite hard to sustain that such variations were due to changes only to the composition of the sediment assessed at each site. Moreover, there are some physiological aspects of specimen used that were not considered, and could explain the variations in the metabolites assessed. For instance reproduction activity, it was not mentioned whether this study was performed during the reproductive or resting period. What was the stage of maturation of the cockles used? The specimens used were females or males? Additionally, such variations in metabolite composition could be due to changes of seawater salinity. Bivalves use organics compounds, mainly aminoacids, to adjust the osmotic pressure with changes of salinity. Seawater salinity certainly changes between the sampling sites, affecting in different way the cockles used for this study. The metabolite composition also could be due to changes in the size and condition of cockles used. All this information was also completely omitted. Finally, authors should be clearly stated in this study, that obtained results are only from a narrow window of time, all this could change between seasons over one-year period. All issues mentioned above, as well the specific comments described below, should be considered and clear stated in the reviewed version of this study. My recommendation is that all these major revisions should be attended by the authors in order to accept this study for publication in this journal.

Specific comments:

Line 9: Use “molluscs” instead of shellfish, this should be done along the text.
Line 14: Should be stated and described in the abstract which metabolites were assessed.
Line 20: Bivalve molluscs are interacted with several environmental factors, not only with sediments. The explanation must be opened to more aspects related to the environment.
Lines 45 and 48: References are written with different font.
Line 62 to 63: Suspended sediment affects seawater turbidly, which in turns could stress bivalve filter feeders. Although could be an important environmental stressor, this could be lesser important for borrowing mollusc, as cockles. This should be considered and widely discussed in this study.
Lines 64 to 66: This is in accordance with mentioned in the general comments. Please, consider it and discuss.
Line 107: Although were considered 8 different sites, at the end 7 sites were only sampled, because some technical issues were occurred. All this is confuse until an explanation came up in the footnote of the area of study. This should clearly state in M&M. There is no reason of add a sampling point which at the end was not sampled. I recommend just omitted.
Line 109: Are those really measurements of health assessment?
Line 121: Why did you boil the samples in ethanol? At which temperature did you boil the samples? Photosynthetic pigments as chlorophyll are quite sensitive prior to oxidation, mainly to light and high temperature.
Line 126 to 127: This should be written in past tense. This should be reviewed and rewritten.
Line 138: The same for specimen sampled. At the end were only 7 sites sampled.
Line 164: You did not assess metabolite expression. This should be reviewed and rewritten.
Line 203: Although were only six metabolites that shown significant differences between sites, I suggest mention the remaining metabolites that were not significant different.

Experimental design

No comment

Validity of the findings

No comment

Additional comments

All is the first part of the reviewing

---

## Round 0.2 · accepted · Accept

· Academic Editor

Accept

Thank you for following through the suggestions and comments from reviewers. The current version of the manuscript is ready for publication.

Reviewer 2 ·

Basic reporting

All comments and suggestions were properly addressed, no further comments.

Experimental design

All comments and suggestions were properly addressed, no further comments.

Validity of the findings

All comments and suggestions were properly addressed, no further comments.

Additional comments

All comments and suggestions were properly addressed, no further comments.